# Mechanisms Operating in the Use of Transition Metal Complexes to Combat Antimicrobial Resistance

**DOI:** 10.3390/microorganisms13071570

**Published:** 2025-07-03

**Authors:** Shiming Wu, Meishu Wang, Ziyi Liu, Chen Fu

**Affiliations:** College of Pharmaceutical Sciences, Southwest University, Chongqing 400715, China; wsi1333@126.com (S.W.); shutong6855@163.com (M.W.); 13668005217@163.com (Z.L.)

**Keywords:** transition metal complexes, drug resistance, bacterial infection, antibacterial mechanism

## Abstract

The increasing diversity and escalating drug resistance of bacterial pathogens have significantly compromised the efficacy of conventional antimicrobial agents, creating formidable challenges in modern infection control. These developments underscore the critical need for innovative therapeutic strategies to address the persistent global health burden posed by microbial resistance. While metal-based compounds have been extensively studied for their anticancer properties in clinical applications, their potential in antimicrobial contexts remains relatively underexplored. This review systematically elaborates on the structure-activity relationship of metal complexes, with a focus on the unique characteristics of metal drugs that differ from organic small molecules. These drugs can overcome drug resistance through various mechanisms (such as generation of reactive oxygen species and penetration of biological membranes). Understanding these mechanisms provides a crucial basis for guiding ligand design and the development of delivery systems.

## 1. Introduction

Transition metal complexes are gaining prominence as strategic antimicrobial candidates to combat the global crisis of microbial resistance, driven by the declining efficacy of conventional antibiotics. Transition metal complexes’ attributes, originally leveraged for catalysis, photoelectric applications, and biocompatible designs, now enable multifactorial mechanisms for microbial disruption. These properties allow cationic species to destabilize bacterial membranes, redox-active surfaces to generate bactericidal reactive oxygen species (ROS), and programmable coordination architectures to target pathogen-specific vulnerabilities [1].

The bioactivity profiles of metal complexes are intrinsically linked to their electronic structures. As evidenced by the CO-ADD dataset (Figure 1), metals with distinct electronic configurations exhibit divergent toxicity profiles: Redox-active metals (e.g., Fe, Cu, Co) frequently display high cytotoxicity due to Fenton-type reactions that generate indiscriminate ROS in mammalian cells [2]. This aligns with CO-ADD data showing no non-toxic hits for iron complexes despite high submission numbers (*n* = 108); d^8^/d^6^ low-spin metals (e.g., Ru(II/III), Ir(III), Pt(II)) demonstrate higher selectivity. Their kinetic inertness (slow ligand exchange) and tunable ligand fields minimize off-target interactions, resulting in >50% non-toxic antimicrobial hits for Ru/Ir systems; soft acid metals (e.g., Ag(I)) exploit bacterial thiophilicity to target thiol-rich proteins, but their lability contributes to mammalian cytotoxicity. CO-ADD data show Ag complexes constitute 17% of active agents, yet only 40% pass non-toxic thresholds. This electronic-toxicity correlation underscores that optimizing metal-centered properties (oxidation state, d-electron count, ligand field strength) is critical for enhancing therapeutic indices.

This review analyzes the antimicrobial mechanisms of transition metal coordination complexes, focusing on membrane disruption, enzyme inhibition, and biofilm penetration through representative metal ligand systems. By correlating structural features (e.g., ligand geometry, oxidation states) with biocidal efficacy, we establish design principles for optimizing antibacterial activity. This work aims to bridge the gap between metallodrug design and microbial pathogenesis, providing a roadmap for next-generation antimicrobials.

## 2. Microbial Infections

Microbial infections represent a critical challenge in clinical medicine, primarily encompassing bacteria, archaea, and fungi. These microorganisms invade host organisms through diverse mechanisms, exploiting host resources for proliferation and potentially triggering immune responses that culminate in disease pathogenesis [3]. Microbial infections, predominantly caused by bacteria and fungi, pose a persistent and escalating global health challenge. Pathogens such as *Staphylococcus aureus* (causing sepsis and pneumonia), *Mycobacterium tuberculosis* (tuberculosis), *Escherichia coli* (urinary tract infections), and *Candida albicans* (invasive candidiasis) exemplify the diversity and severity of these threats [4,5,6]. The efficacy of conventional antibiotics against these pathogens is increasingly compromised by the rise of antimicrobial resistance (AMR).

The inappropriate and excessive use of antibiotics in human medicine and agriculture drives bacterial resistance to antibacterial agents. This has led to the emergence of high-priority drug-resistant pathogens such as MRSA (methicillin-resistant *Staphylococcus aureus*) and VRE (vancomycin-resistant Enterococcus). Based on the data analysis from previous studies, in 2019, there were 4.95 million deaths worldwide related to bacterial AMR, among which 1.27 million deaths were directly caused by bacterial AMR. By 2050, the annual mortality rate directly caused by AMR is expected to rise to 10 million, with the highest estimated number of deaths in Asia, followed by Africa. This is mainly due to the large population and the lack of regulation related to AMR prevention, which has now become a serious healthcare safety issue [7]. As shown in Figure 2, by 2023, a global map of antibiotic resistance due to antibiotic usage had been collected (Figure 2). Among them, the regions with the highest antibiotic usage rates are the more developed areas such as North America, Europe, and Australia. Antibiotics are also used in some parts of Africa and Asia [8].

During the process of bacterial transmission and treatment, bacteria have evolved a series of defense mechanisms to avoid or mitigate the damage caused by drugs. The main resistance mechanisms include drug efflux pump systems (which expel drugs to reduce intracellular concentrations), plasmid-mediated gene transfer (the exchange of resistance genes through mobile genetic elements such as plasmids), and biofilm formation (dense biological films that hinder drug penetration) among others [9]. Common drug-resistant bacteria include Methicillin resistant Staphylococcus aureus (MRSA), Vancomycin-resistant Enterococci (VRE), Carbapenem-resistant Enterobacteriaceae (CRE), and multidrug-resistant Mycobacterium tuberculosis (MDR-TB) [10]. MRSA resists β-lactam antibiotics by producing a structurally abnormal penicillin-binding protein PBP2a (also known as PBP2′). This protein retains the function of bacterial cell wall synthesis while its affinity for β-lactam drugs (such as methicillin) is significantly reduced due to the change in spatial conformation, preventing the drugs from effectively inhibiting cell wall synthesis and thus causing drug resistance [11]. VRE acquires acquired resistance genes (such as vanA and vanB) to chemically modify the D-alanyl-D-alanine (D-Ala-D-Ala) at the terminal end of the cell wall precursor, transforming it into D-alanyl-D-lactic acid (D-Ala-D-Lac). This structural change directly disrupts the hydrogen bond binding between vancomycin and the target, resulting in a decrease in antibiotic affinity and failure [12]. CRE mainly hydrolyzes drugs such as imipenem and meropenem by producing carbapenemases (such as KPC type, NDM type, or OXA-48 type). These enzyme genes are often carried by plasmids and can spread rapidly between different bacterial strains through horizontal gene transfer, posing a serious challenge to clinical treatment [13]. The MDR-TB mechanism of Mycobacterium tuberculosis is completely different from the above bacteria; its resistance mainly results from chromosomal gene mutations. For example, a mutation in the katG gene will cause the inactivation of catalase-peroxidase, preventing isoniazid (INH) from being activated into toxic free radicals, while a mutation in the rpoB gene (such as S450L) will change the conformation of the β-subunit of RNA polymerase, blocking the binding of rifampicin (RIF) to the target, ultimately leading to the failure of both these first-line anti-tuberculosis drugs and resulting in a highly difficult-to-treat multi-drug resistant tuberculosis [14].

Drug-resistant bacteria pose a challenge to global public health, and coordinated measures such as rational use of antibiotics, enhanced infection control, and antimicrobial stewardship are needed. Similarly, antifungal drug resistance has become a key threat to clinical patients, and its cause lies in the overuse of antifungal drugs. The development of new antibacterial and antifungal drugs is crucial in combating bacterial and fungal resistance.

The escalating threat of antimicrobial resistance (AMR) poses a global public health crisis, the increase in antimicrobial resistance caused by excessive use of antimicrobial drugs has become an extremely serious challenge in clinical settings, especially for patients with compromised immune systems. Addressing these dual threats requires urgent innovation in drug development, as traditional therapeutics lose efficacy against pathogens armed with sophisticated resistance mechanisms. This review explores recent advances in transition metal-based agents, a promising class of antimicrobials designed to circumvent resistance by targeting novel pathways.

## 3. Applications of Transition Metal Complexes in Antimicrobial Fields

Transition metal complexes (e.g., silver, copper, platinum, iridium, and ruthenium) have demonstrated significant potential in antimicrobial applications. These complexes combine the properties of metal centers and organic ligands, enabling them to inhibit or kill bacteria and fungi through multiple mechanisms. Below, we highlight the applications and research progress of several transition metal complexes in antimicrobial fields.

### 3.1. Silver Complexes

Silver has been utilized for its antimicrobial properties since antiquity. For example, the ancient Romans employed silver containers to store water and food to prevent spoilage. In modern medicine, silver sulfadiazine (AgSDZ) as a topical agent for burn wound infections [15]. Structure and mechanism of action: its structure consists of a polymeric network wherein silver(I) ions coordinate with sulfadiazine anions via the pyrimidine nitrogen and the deprotonated sulfonamide nitrogen, forming argentophilic interactions. This unique configuration contributes to its extremely low aqueous solubility (<2.8 μmol/L at pH 7) and controlled release of bioactive Ag^+^ ions in biological environments.

AgSDZ primarily functions through dissociation into Ag^+^ and sulfadiazinate upon contact with biological media (e.g., serum, electrolytes). Key mechanisms include the following: DNA binding, where Ag^+^ binds to bacterial DNA and inhibits replication and multi-target effects, including interactions with Cl^−^, amino acids, and proteins (e.g., serum albumin) that modulate silver bioavailability and toxicity. The antibacterial activity of AgSDZ exhibits broad-spectrum activity against both Gram-positive bacteria (e.g., Staphylococcus aureus, MIC: 50–280 μmol/L) and Gram-negative bacteria (e.g., Pseudomonas aeruginosa, MIC: 25–140 μmol/L), though efficacy varies with bacterial strain, solvent (DMSO suspensions yield higher MICs), and exposure time [16].

Recent advances include novel silver(I)-sulfonamide complexes to address bacterial resistance, with recent efforts focus on designing new Ag(I)-sulfonamide complexes: sulfadoxine-Ag, with 300× enhanced antifungal activity against Candida albicans (MIC: 3.5 μmol/L) compared to the free ligand; Nimesulide-Ag, which exhibits selective antitumor activity against melanoma (UACC-62, TGI: 2.8 μmol/L) and in vivo tumor regression in SCC models; and heteroleptic complexes, with improved potency against *P. aeruginosa* (MIC: 2.31 μmol/L) via thiocyanate/phenanthroline ancillary ligands. While AgSDZ laid the foundation for medicinal silver chemistry, next-generation silver-sulfonamide complexes show promise for targeted antibacterial, antitumor, and antiviral applications. Future work should address speciation in biological systems and challenges in clinical translation [15,17].

### 3.2. Copper Complexes

Copper, a transition metal, has been employed as an antimicrobial agent since antiquity due to its nonspecific targeting capabilities against microorganisms. In recent years, copper’s remarkable affinity for biological ligands and redox properties have positioned it as an ideal metal for biochemical reactions [18]. Copper complexes formed through coordination bonds between Cu^2+^/Cu^+^ ions and organic ligands exhibit potent antimicrobial effects through multiple mechanisms. For instance, 8-aminoquinoline-5-nitrouracil-copper (Figure 3, Complex **1**) represents the first copper-based complex effective against Plesiomonas shigelloisis, where metal-mediated structural optimization enhances its ability to inhibit DNA gyrase and topoisomerase IV, outperforming its parent ligand [19].

Another notable example, the phenanthroline-quinolone hybrid complex Cu-DPPZ-CipA (Figure 3, Complex **2**), demonstrates dual DNA-targeting mechanisms. Upon Cu (II) binding, this complex generates reactive oxygen species (ROS) via Fenton-like reactions while simultaneously intercalating into the DNA’s minor groove. The combined effects induce oxidative DNA damage and structural destabilization, significantly enhancing its antibacterial potency against Gram-negative pathogens [20].

Beyond quinolone derivatives, copper complexes targeting other topoisomerase isoforms have shown promise. [Cu(N,N,S-py2tsc-N1HR2)X] (X = I, Cl, Br) (Figure 3, Complex **3**) inhibits topoisomerase II and exhibits potent anti-biofilm activity; 100% bactericidal efficacy was achieved against *Staphylococcus aureus*, *Salmonella enteritidis*, and *Klebsiella pneumoniae*. This complex disrupts bacterial replication by stabilizing DNA-topoisomerase cleavage intermediates, offering a strategic approach to combat drug-resistant strains [21].

In addition to direct antimicrobial action, certain copper complexes address resistance mechanisms. The binuclear complex [Cu(2,2-bipy)_2_(C_3_H_3_O_2_)_6_(H_2_O)] (Figure 3, Complex **4**) acts as an efflux pump inhibitor, as confirmed by flow cytometry. By blocking bacterial drug-efflux systems, it synergizes with conventional antibiotics to overcome resistance. Notably, this complex maintains cytocompatibility with mammalian cells and avoids inducing oxidative stress, underscoring its potential for clinical applications in combination therapies [22].

Recent advances include binuclear Cu(II) complexes **5a/5b** (Figure 3, Complex **5a/5b**), synthesized from Schiff base ligands HL1 (N′-(1-(2-hydroxyphenyl) ethylidene)-2-phenylacetohydrazide) and HL2 (N′-((1-hydroxyna-phthalen-2-yl) methylene)-2-phenylacetohydrazide). These complexes exhibit broad-spectrum antimicrobial efficacy matching clinical standards (ampicillin/amphotericin B), with 2× higher antifungal activity against *C. albicans* than earlier analogs. Their dual mechanism combines enhanced membrane penetration (via reduced Cu^2+^ polarity) and ROS generation through Cu(I)/Cu(II) redox cycles. Additionally, **5a** demonstrates potent antioxidant activity, positioning both complexes as multifunctional agents against drug-resistant pathogens [23].

### 3.3. Platinum Complexes

Platinum complexes are one of the most successful metal-based drug categories in cancer treatment. Cisplatin, carboplatin, and oxaliplatin are established anticancer agents, while recent studies have highlighted the expanding potential of platinum complexes in combating microbial infections, with structural modifications playing a pivotal role in enhancing their antimicrobial efficacy [24]. Pt(II) complexes incorporating electron-withdrawing groups such as nitro (-NO_2_) and fluorine (-F) demonstrate significantly improved DNA-binding capacity and antimicrobial activity compared to their unmodified counterparts. This enhancement arises from specific interactions like O-H bond formation within thymidine heterocyclic compounds, which strengthen DNA adduct formation and boost pathogen inhibition [25]. The structural diversity of these complexes, particularly through the use of bridging ligands, enables the formation of cyclometalated complexes that exhibit stronger DNA binding and superior antimicrobial effects relative to mononuclear analogs [26]. Conversely, electron-donating substituents like -OCH_3_ and -CH_3_ have been shown to reduce activity against *Pseudomonas aeruginosa* and *Escherichia coli* [27].

The nuclearity of platinum complexes critically influences their antimicrobial performance. Binuclear Pt(II) complexes, exemplified by chloride-bridged [Pt_2_LCl_4_] structures, penetrate bacterial membranes through chelation-mediated mechanisms, with substituent-induced toxicity dictating their antimicrobial effects (Figure 4, Complex **6**). Fluorinated heterocyclic ligands, such as 5-perfluoroalkyl-1,2,4-oxadiazolylpyridine, further enhance activity against *E. coli* and *S. aureus* by forming exceptionally stable DNA adducts [28]. The presence of multiple metal centers in these complexes facilitates the creation of intricate DNA crosslinking patterns, significantly improving antimicrobial efficiency and potentially overcoming drug-resistant infections through novel damage mechanisms [29]. Parallel research on N-substituted imidazolium salts reveals that antimicrobial activity is profoundly influenced by alkyl chain length, with complexes containing 8–16 carbon atoms achieving the lowest minimum inhibitory concentrations. This phenomenon correlates with the ability of long hydrocarbon chains to integrate into bacterial membrane lipid bilayers, disrupting membrane integrity and inducing cellular content leakage [30,31,32].

Complementary work by Al-Khathami et al. on platinum(II) Schiff base complexes demonstrates structure-activity relationships in antimicrobial applications (Figure 5, Complex **7**) [33]. Using pyridine-2-carboxaldehyde-derived ligands with various aromatic amines, these ligands exhibit significant inhibitory effects on a variety of pathogens including *E. coli*, *S. aureus*, and *Candida*. Complexes bearing electron-withdrawing groups (particularly nitro substituents) exhibited superior DNA binding and broad-spectrum antibacterial activity, though fungal inhibition remained challenging. Intriguingly, free ligands containing hydroxyl groups showed enhanced activity against Gram-negative bacteria compared to their platinum-bound counterparts, suggesting competing mechanisms of action between the ligand and metal center.

The biological activity of platinum complexes fundamentally stems from their DNA interaction mechanisms, which involve three critical phases: the pre-binding of the DNA, the formation of adducts, and the conformational changes of the DNA [34]. While traditional mononuclear agents like cisplatin form specific DNA crosslinks, multinuclear platinum complexes introduce a paradigm shift by containing two or more platinum centers capable of creating structurally distinct DNA adducts. These multi-nuclear systems exhibit enhanced DNA-binding versatility through bridged halogen ligand lability; our prior research demonstrated that bridging bonds in polynuclear complexes display greater reactivity than terminal bonds, enabling their controlled dissociation into bioactive monomers via aquation processes upon entering biosystems [35].

The strategic design of P,N- and S,N-type bidentate further expands this functionality by combining soft/hard base properties to orchestrate metal coordination geometry (Figure 6). These ligands not only direct the spatial organization of metal centers but also facilitate the construction of bimetallic or polynuclear architectures [36]. Figure 6 specifically illustrates how such ligands mediate DNA-drug interactions: (1) the soft sulfur/nitrogen sites stabilize metal coordination, while (2) the hard phosphorus/nitrogen donors enable directional DNA backbone binding, collectively inducing helical distortion that disrupts replication machinery.

This multilayered DNA targeting approach synergizes with insights from other metallodrug classes. For instance, while gold complexes employ thiol-enzyme inhibition to trigger oxidative apoptosis [37], platinum systems leverage nuclearity-controlled DNA adduct diversity. Future development requires optimizing both electronic modulation (electron-withdrawing/donating substituents) and nuclearity engineering (mono- vs. multinuclear architectures) to design metallodrugs that concurrently address antimicrobial resistance and cancer evolution through precision DNA damage and epigenetic modulation.

### 3.4. Ruthenium Complexes

Hexacoordinate ruthenium complexes have emerged as a research hotspot in chemical biology and medicinal chemistry due to their excellent physiological stability, versatile photophysical and electrochemical properties, variable oxidation states, and biomolecular binding activity [38]. By modifying the metal center, complexes with similar geometric parameters but distinct redox and kinetic characteristics can be designed. The chemical inertness of ruthenium complexes allows precise structural tuning to achieve stereoelectronic complementarity with target active sites.

Ruthenium complexes exhibit both anticancer and antimicrobial properties. For the most representative polypyridyl ruthenium complex, tris(2,2′-bipyridyl)-ruthenium(II) complex [Ru[(bpy)_3_]^2+^ (Figure 7, Complex **8**), there are many derivatives and it is widely used in photooxidation-reduction catalysis and the life science fields [39]. [Ru(phen)_3_]^2+^ (Figure 7, Complex **9**) serves as a foundational structure, composed of three 1,10-phenanthroline (phen) ligands [40]. While inactive against Gram-positive, Gram-negative, and acid-fast bacteria, its methyl-substituted derivatives [Ru(phen)_2_(acac)]^+^ (Figure 7, Complex **10**) demonstrated dramatically enhanced activity. [Ru(Me_4_phen)_2_(acac)]^+^, incorporating 3,5,6,8-tetramethyl phenanthroline (Me_4_phen) and acetylacetonato (acac) ligands, exhibited potent inhibition against Staphylococcus pyogenes, with only a two-fold increase in bacterial resistance and a stark contrast to the 10,000-fold resistance observed with penicillin. This marked the first evidence of low-resistance potential in metal complexes. [Ru(2,9-Me_2_phen)_2_(dppz)]^2+^ (Figure 7, Complex **11**), combining 2,9-dimethylphenanthroline and the DNA- intercalating dipyrido [3,2-a:2′,3′-c] phenazine (dppz) ligand, achieved a remarkably low MIC of 2 μg/mL against methicillin-resistant strains and reduced eukaryotic toxicity in Caenorhabditis elegans infection models, revealing a synergistic mechanism of DNA targeting and lipophilicity [41].

Mechanistic studies revealed that [Ru(phen)_2_(p-BPIP)]^2+^ (Figure 7, Complex **12**) induces DNA fragmentation and RNA degradation in *M. tetragenus*, which is indicative of necrotic cell death. These effects were concentration-dependent, with pronounced RNA smearing observed at 15 μM of the complex. SEM imaging further demonstrated that Complex **12** disrupts bacterial membrane integrity, increasing permeability and leading to cell death. The antibacterial mechanism of this ruthenium complex centers on DNA and RNA damage, aligning with behaviors observed in related metal agents. For instance, such complexes primarily target nucleic acids, highlighting a common mode of action among this class of agents [42].

Half-sandwich Ru(II) arene complexes have emerged as promising antifungal agents with tunable specificity. The archetypal complex [Ru(*η^6^*-p-cymene)Cl_2_(pta)] (pta = 1,3,5-triaza-7-phosphatricyclo [3.3.1.1] decane) (Figure 8, Complex **13**) demonstrated non-toxic growth inhibition against Trichophyton mentagrophytes (MIC = 8.2 μM) through mechanisms distinct from general cytotoxicity [38]. Structural optimization via ligand design significantly enhanced potency: replacement of pyridine with benzothiazole in N,S-thiourea derivatives yielded sub-10 nM MIC values against *C. albicans* (6.77 nM for Figure 8, Complex **14**) and *C. neoformans* (4.83 nM for Figure 8, Complex **15**). Notably, antifungal efficacy is inversely correlated with ligand multiplicity in azole-based complexes (Figure 8, **16**–**18**), where tris-azole species showed reduced activity despite maintaining 1.5–2.8-fold growth suppression at 0.01–0.5 mM. Crucially, these complexes bypass classical ergosterol biosynthesis inhibition, as their N3 coordination sites are occupied by ruthenium, suggesting novel targets such as membrane integrity disruption or redox modulation. This ligand-driven specificity, combined with low mammalian cytotoxicity (IC_50_ > 100 μM in HepG_2_ cells), positions half-sandwich Ru(II) architectures as versatile platforms for combating drug-resistant fungal pathogens [43].

Hexacoordinate polypyridyl Ru(II) complexes demonstrate versatile antimicrobial potential through rational ligand engineering. Hydrazide-functionalized derivatives (Figure 9, Complex **19**–**21**), featuring polypyridyl cores modified with -NH–NH_2_ groups, exhibit enhanced lipophilicity via chelation, enabling efficient penetration of bacterial and fungal membranes [44]. Their antimicrobial activity is concentration-dependent, with MIC values decreasing 3–5× at elevated concentrations (11–13 μM), and shows selectivity toward Gram-positive pathogens like *Staphylococcus aureus*, likely due to ribosomal structural disparities or membrane permeability variations [45]. In contrast, the dinonyl-bipyridine [(phen)_2_Ru(bpy-dinonyl)_2_] (Figure 9, Complex **22**) targets fungal infections, particularly *C. neoformans*, with superior efficacy compared to fluconazole (MIC = 1.6 vs. 4.1 μg/mL; MFC = 16.2 vs. 64.4 μg/mL). Its hydrophobic 4,4′-dinonyl-2,2′-bipyridine ligands enhance membrane disruption and ROS generation, while RDP peptide-conjugated liposomes facilitate blood-brain barrier traversal for treating cryptococcal meningitis. Notably, Complex **22** integrates therapeutic and diagnostic functions, emitting red fluorescence to track DNA conformational changes under pH modulation and exemplifying theranostic duality. These Ru(II) architectures highlight the critical role of ligand hydrophobicity in optimizing antimicrobial penetration and the translational promise of combining targeted delivery with real-time imaging [46].

### 3.5. Iridium Complexes

Iridium metal complexes exhibit multiple oxidation states (+1, +3, +4), enabling the formation of diverse complexes. Their strong coordination capacity and excellent photophysical properties (luminescence) make them widely applicable in photodynamic therapy and bioimaging. By introducing varied organic ligands (e.g., pyridine, bipyridine, and phenanthroline), the targeting specificity and multifunctionality of the complexes can be modulated.

While reports on the antimicrobial activity of organometallic iridium(III) complexes remain limited, key studies highlight their unique mechanisms and therapeutic potential. Keene and Collins (2013) pioneered comparative analyses of Ru(II) and Ir(III) complexes, revealing that Ir(III) derivatives (overall charge +4 post-aquation) exhibit bacteriostatic rather than bactericidal effects against Gram-positive and Gram-negative bacteria [43]. This reduced efficacy was attributed to hindered membrane penetration due to higher charge density and the lability of chloride counterions, which impair bacterial accumulation. However, subsequent work on cyclometalated Ir(III)-polypyridyl complexes demonstrated their ability to interact with DNA via intercalation or act as photosensitizers, generating reactive oxygen species (ROS) under light irradiation to induce oxidative damage [47]. A landmark example is ([Ir(Cp*)(biguanide) Cl]^+^ (Cp* = pentamethy lcyclopentadienyl; biguanide = metformin derivative) (Figure 10, Complex **23**), developed by Chen et al. The complex showed broad-spectrum efficacy with MICs of 0.125 μg/mL against MRSA (4× more potent than vancomycin) and 0.25 μg/mL against *C. albicans* and *C. neoformans* (32× superior to fluconazole), while synergistically restoring vancomycin activity against resistant *Enterococci* and disrupting > 70% of *S. aureus* biofilms at sub-MIC levels, though limited against *Pseudomonas aeruginosa* (MIC > 32 μg/mL) due to efflux-mediated resistance [47].

In our laboratory, a series of polypyridyl iridium(III) complexes (Figure 11, Complexes **24**–**28**) were developed. Among these, the complex [(ptpy)_2_Ir(dppz)]PF_6_ demonstrated notable broad-spectrum antifungal activity, with MIC values of 1–8 μg/mL against drug- resistant pathogens and low hemolytic activity toward mammalian cells. This complex also exhibited biofilm inhibition and disruption capabilities, with a low propensity to induce drug resistance. In contrast, [Ru(phen)_2_(dppz)]^2+^ showed weaker antifungal activity and higher hemolytic activity. The antibacterial mechanism of iridium complexes is likely linked to DNA binding, with activity differences attributed to variations in chemical structure and charge.

Recently developed polypyridyl iridium(III) complexes (Figure 11, Complexes **24**–**28**) revealed significant antimicrobial potential, with the leader being Complex **26** ([(ptpy)_2_Ir(dppz)]PF_6_, Figure 11), demonstrating broad-spectrum antifungal activity against drug-resistant pathogens (MIC = 1–8 μg/mL, Table 1) and low hemolytic toxicity (HC_50_ > 256 μg/mL) [45]. Unlike the narrower spectrum [Ru(phen)_2_(dppz)]^2+^, which showed negligible activity against *C. albicans*, despite high hemolysis, Complex **26** additionally inhibited biofilm formation and exhibited low resistance propensity. The enhanced efficacy of Complex **26** is attributed to its distinct chemical architecture and charge distribution, though both complexes likely share DNA-binding antimicrobial mechanisms.

Iridium(III) complexes have emerged as dual-targeting antimicrobial agents with multifunctional therapeutic potential, synergizing with conventional antibiotics while evading resistance mechanisms through structural and mechanistic innovations. Their low mammalian cytotoxicity (IC_50_ > 100 μM in HEK293 cells) underscores high therapeutic selectivity, making them promising candidates for treating drug-resistant infections. Future designs could optimize ligand hydrophobicity to counteract efflux pump-mediated resistance in Pseudomonas aeruginosa or incorporate photoresponsive ligands to enable spatiotemporal activation of reactive oxygen species (ROS), leveraging Ir(III)’s inherent photophysical advantages for precision therapy. These complexes also exhibit biofilm disruption capabilities (>70% inhibition at sub-MIC levels), offering a strategic solution to persistent infections. By engineering ligands (e.g., bipyridine, therapeutic efficacy) to enhance the targeting specificity and photodynamic efficacy, Ir(III) architectures can be tailored as next-generation antimicrobials that combine real-time bioimaging, light-controlled activity, and resistance mitigation.

Transition metal complexes-silver, copper, platinum, ruthenium, and iridium-exhibit broad-spectrum antimicrobial efficacy against critical pathogens highlighted in this review, including drug-resistant bacteria (MRSA and VRE), invasive fungi (*C. albicans*, *C. neoformans*, *Aspergillus*), and persistent biofilms. Silver nanoparticles disrupt Gram-negative membranes (e.g., *E. coli*, *V. cholerae*) via ROS generation, while copper complexes target DNA gyrase in *P. shigelloides* and *S. aureus*. Platinum architectures can overcome multidrug-resistant *M. tuberculosis* and *P. aeruginosa* through DNA cross-linking, whereas ruthenium and iridium systems penetrate fungal biofilms (e.g., *C. albicans*, *C. neoformans*) through photodynamic ROS and nucleic acid intercalation. Critically, these complexes counter resistance mechanisms prevalent in XDR/PDR pathogens-efflux pumps, enzymatic inactivation, and peptidoglycan remodeling-via multimodal mechanisms (membrane destabilization, redox cycling, enzyme inhibition). Future designs must prioritize pathogen-specific vulnerabilities, such as *Aspergillus* melanin biosynthesis or MRSA virulence factor suppression, while integrating biocompatible delivery platforms to address systemic infections (e.g., cryptococcal meningitis and invasive candidiasis). By deeply integrating structural tunability with the research on microbial pathogenic mechanisms, Transition metal complexes offer multi-dimensional solutions to the increasingly severe threat of antibiotic resistance.

### 3.6. Comparative Summary of Antimicrobial Metal Complexes

A comparative analysis of the key characteristics, antimicrobial mechanisms, efficacy profiles, and limitations of the transition metal complexes discussed in this review is presented in Table 2. Building upon the mechanistic insights detailed in Section 3.1, Section 3.2, Section 3.3, Section 3.4 and Section 3.5, Table 2 provides a consolidated analysis of the key characteristics, antimicrobial mechanisms, and limitations of transition metal complexes. This comparative framework highlights how distinct metal centers and ligand architectures dictate biological activity against priority pathogens while underscoring shared translational challenges.

## 4. Challenges and Future Perspectives

While transition metal complexes exhibit significant antimicrobial potential, their clinical translation faces critical challenges including systemic toxicity from nonspecific metal accumulation in host tissues (e.g., nephrotoxicity of Pt complexes) and hemolytic effects of cationic species (e.g., Ru polypyridyls). Current strategies to mitigate these issues focus on targeted delivery systems (e.g., nanoparticles) to reduce off-target toxicity while enhancing pathogen-specific accumulation. Concurrently, emerging resistance mechanisms against metal-based agents-such as efflux pumps (SilE/SilP for Ag^+^) and metalloregulatory mutations (for Cu^2+^)-necessitate combinatorial approaches with conventional antibiotics (e.g., Ir-MFC synergized with vancomycin against resistant Enterococci) to delay resistance evolution. Bioavailability limitations, particularly the poor aqueous solubility of hydrophobic complexes (e.g., Ir-Cp*), are being addressed through structural modifications using polar ligands (glycosylation) or nanoformulations (RDP peptide-conjugated Ru liposomes). Furthermore, mechanistic ambiguities arising from indirect evidence (e.g., ROS detection) underscore the need for advanced techniques like real-time imaging leveraging the luminescent properties of Ir/Ru complexes to map spatiotemporal metal distribution and target engagement within pathogens.

Looking ahead, research priorities center on precision antimicrobial design: (1) exploiting pathogen-specific vulnerabilities via microbial metallobiology (e.g., Fe-scavenging pathways in *P. aeruginosa* and melanin biosynthesis in Aspergillus); (2) optimizing therapeutic indices by balancing ligand hydrophobicity (membrane penetration) and hydrophilicity (renal clearance), exemplified by electron-withdrawing groups (-NO_2_, -F), enhancing DNA binding while reducing Pt toxicity, and alkyl chain tuning in Cu-thiosemicarbazones, maximizing membrane integration versus cytotoxicity; (3) and developing stimuli-responsive activation systems (e.g., photoactivatable Ru/Ir complexes) for spatiotemporal ROS generation at infection sites. These integrated advances will drive the development of next-generation antimicrobials with enhanced efficacy, reduced off-target effects, and minimized resistance development.

## 5. Conclusions

Transition metal antimicrobials exhibit diverse antimicrobial strategies, such as ROS-mediated membrane disruption by Ag-NPs, DNA gyrase inhibition via copper complexes, and DNA cross-linking reaction driven by nucleophilic substitution in the platinum-based system. However, their clinical translation remains hindered by nonspecific toxicity and ambiguous targets, which are critical issues that demand resolution for therapeutic advancement.

This review establishes that decoding metal complex mechanisms-from electronic configurations to ligand geometries-is the foundation for unlocking their therapeutic potential and guiding strategic development. Despite these challenges, their intrinsic stereochemical versatility-arising from the interplay between metal centers (e.g., Ru(II)/Ir(III)) and polydentate ligand architectures (e.g., pyridinedicarboxylic acid derivatives)-endows these complexes with multifunctional capabilities. This chemical adaptability facilitates synergistic antimicrobial action: silver species compromise biofilm integrity through electrostatic interactions, ruthenium systems exploit ligand hydrophobicity for fungal biofilm penetration, and iridium derivatives disrupt microbial redox homeostasis. Such multi-modal functionality, encompassing oxidative stress induction, efflux pump suppression, and membrane destabilization, inherently mitigates resistance development while addressing the mechanistic constraints of traditional therapeutic approaches.

Innovative strategies are addressing translational barriers. Third-generation platinum drugs, like Satraplatin, employ oral administration with cyclohexylamine ligands, reducing nephrotoxicity by 70% compared to Cisplatin, while maintaining efficacy against carbapenem-resistant Enterobacteriaceae. Additionally, ligand-stabilized Cu(II) thiosemicarbazone complexes demonstrate dual targeting of *V. cholerae* topoisomerases and membrane phospholipids, achieving synergistic bactericidal effects.

Current research prioritizes elucidating structure-driven pathogen specificity, exemplified by Ru(II)-dppz intercalators selectively degrading *M. tuberculosis* RNA and Ir(III)-Cp* complexes inhibiting the pathogenicity of *Candida krusei* by generating ROS and suppressing DNA damage. Future work must bridge these mechanistic insights with translational engineering (e.g., stimuli-responsive activation) to address clinical barriers, particularly given the divergence between in vitro efficacy against MDR/XDR pathogens (e.g., vancomycin-resistant *Enterococci*) and in vivo outcomes due to metabolic sequestration or off-target accumulation.

Future advances will integrate computational ligand design and multi-omics to decode metal-pathogen interactions, enabling precision-engineered antimicrobials through oxidation state dynamics and ligand stability. Targeting vulnerabilities like efflux pumps or biofilm matrices, structure-guided engineering, and responsive delivery systems will enhance specificity against resistant Mycobacterium tuberculosis, Candida biofilms, and pan-resistant Gram-negative pathogens. Collaborative synergy across chemistry, biology, and clinical science will yield pathogen-adaptive metallodrugs, transforming antimicrobial stewardship to counter global resistance crises.

## Figures and Tables

**Figure 1 microorganisms-13-01570-f001:**
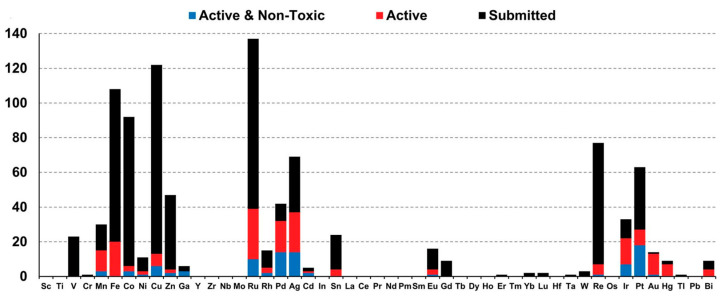
Correlation of metal complex distribution and bioactivity profiles (CO-ADD Dataset): All compounds (black, *n* = 906); active antimicrobial agents (red, MIC ≤ 16 μg mL^−1^ or 10 μM, *n* = 246/906) (quantified by minimum inhibitory concentration, MIC); and active and non-toxic compounds (blue, *n* = 88/246) [2].

**Figure 2 microorganisms-13-01570-f002:**
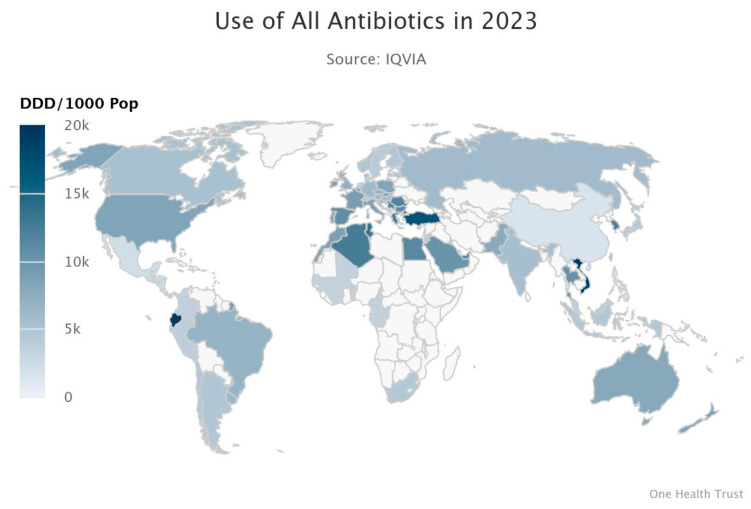
Resistance Map showing the Global usage of antibiotics in 2023 [8].

**Figure 3 microorganisms-13-01570-f003:**
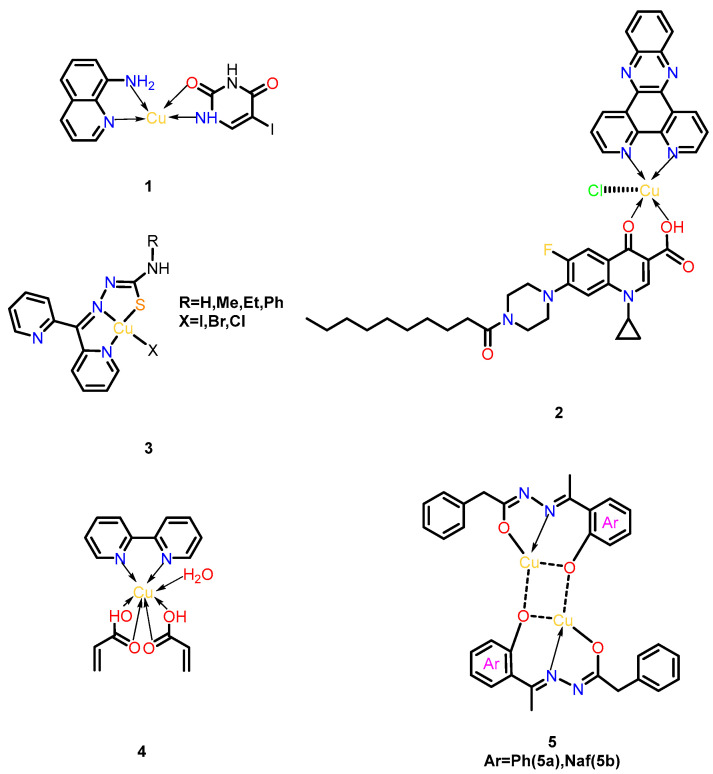
Structural series of copper complexes. (**1**) 8-aminoquinoline-5-nitrouracil-copper; (**2**) Cu-DPPZ-CipA; (**3**) [Cu(N,N,S-py2tsc-N1HR2)X] (X = I, Cl, Br); (**4**) [Cu(2,2-bipy)_2_(C_3_H_3_O_2_)_6_(H_2_O)]; (**5**) Cu(II)(N′-(1-(2-oxyphenyl) ethylidene)-2-phenylacetohydrazide) **5a** and Cu(II)(N′-((1-oxyna- phthalen-2-yl) methylene)-2-phenylacetohydrazide) **5b**.

**Figure 4 microorganisms-13-01570-f004:**
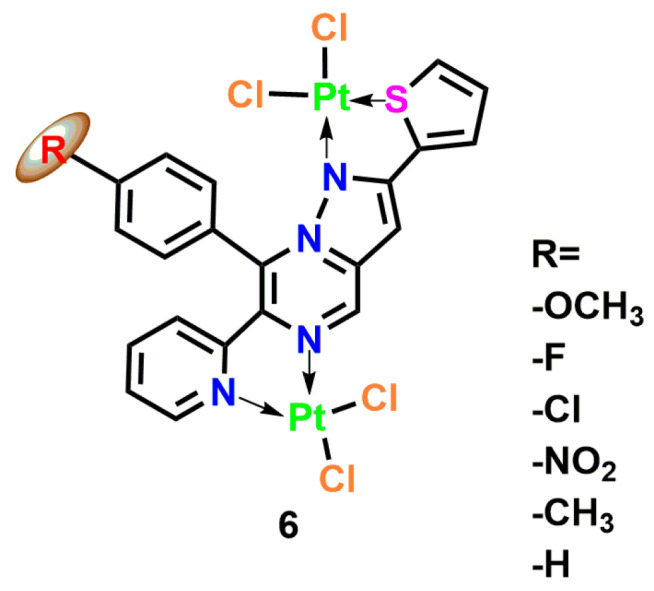
Coordination structure of the binuclear Pt(II) complex [Pt_2_(μ-Cl)_2_(L-R)Cl_2_], where L-R is a tetradentate pyrazolo [1,5-a] pyrimidine ligand with variable substituent R at the para-position of the C-8 phenyl ring (R = -OCH_3_, -F, -Cl, -NO_2_, -CH_3_, -H) Complex **6** [26].

**Figure 5 microorganisms-13-01570-f005:**
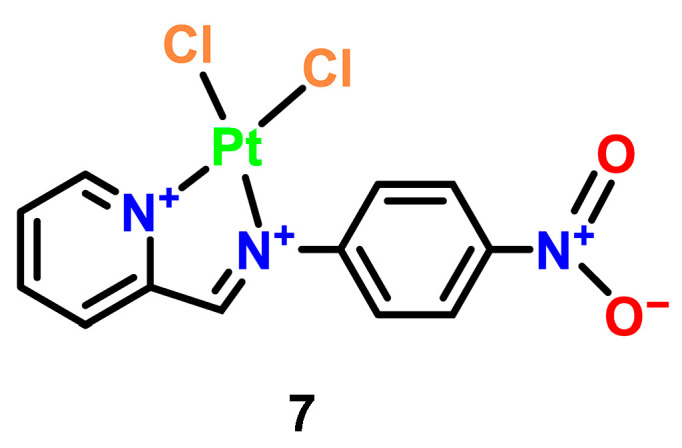
Structural representation of a Pt(II) complex with a 2-(4-nitrophenyl) imidazopyridinedione ligand Complex **7** [28].

**Figure 6 microorganisms-13-01570-f006:**
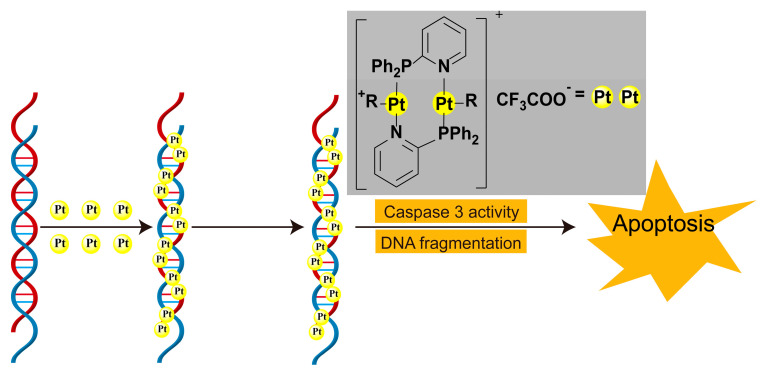
Formation of multi-core platinum(II)-DNA adducts and the pathway of apoptosis in cells [28].

**Figure 7 microorganisms-13-01570-f007:**
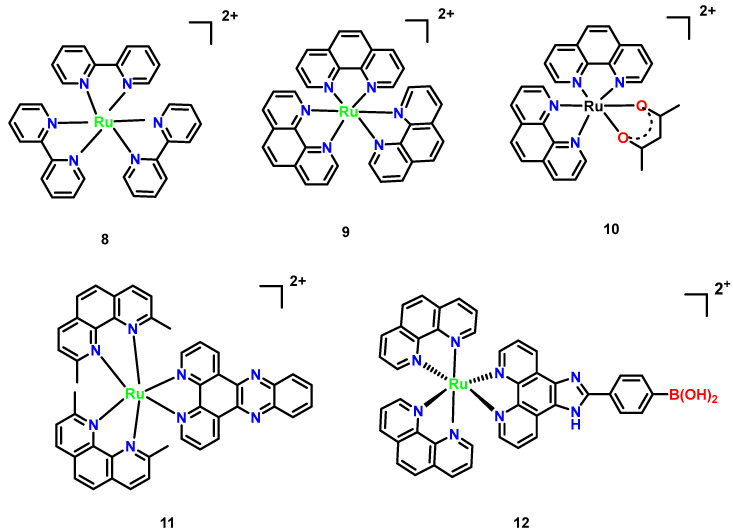
Mononuclear polypyridylruthenium (II) complexes exhibiting antimicrobial activity. (**8**) [Ru(bpy)_3_]^2+^; (**9**) [Ru(phen)_3_]^2+^; (**10**) [Ru(phen)_2_(acac)]^+^; (**11**) [Ru(2,9-Me_2_phen)_2_ (dppz)]^2+^; and (**12**) [Ru(phen)_2_(p-BPIP)]^2+^ [42].

**Figure 8 microorganisms-13-01570-f008:**
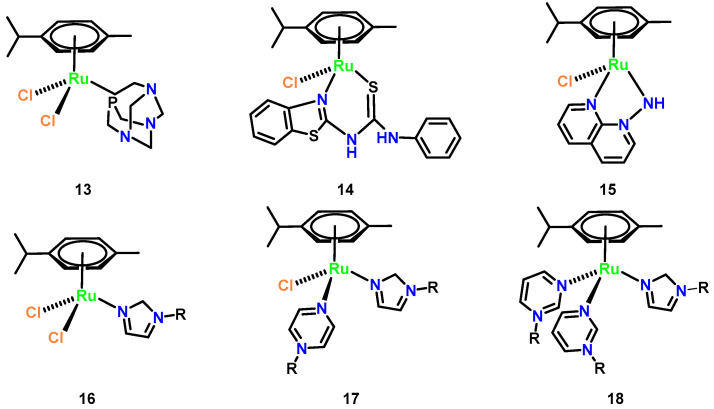
Chemical structures of representative half-sandwich ruthenium(II) arene (Complexes **13**–**18**). (**13**) [Ru(*η^6^*-p-cymene)Cl_2_(pta)]; (**14**) [Ru(*η^6^*-p-cymene)Cl_2_ (Lbenz)]; (**15**) [Ru(*η^6^*-p-cymene)Cl(1,8-naphthyridin-1-amine)]; (**16**) [Ru(*η^6^*-p-cymene)Cl_2_ (imidazole-R)]; (**17**) [Ru(*η^6^*-p-cymene)Cl(pyrazine-R)(imidazole-R)]; and (**18**) [Ru(*η^6^*-p-cymene) (pyrimidine-R)_2_(imidazole-R)].

**Figure 9 microorganisms-13-01570-f009:**
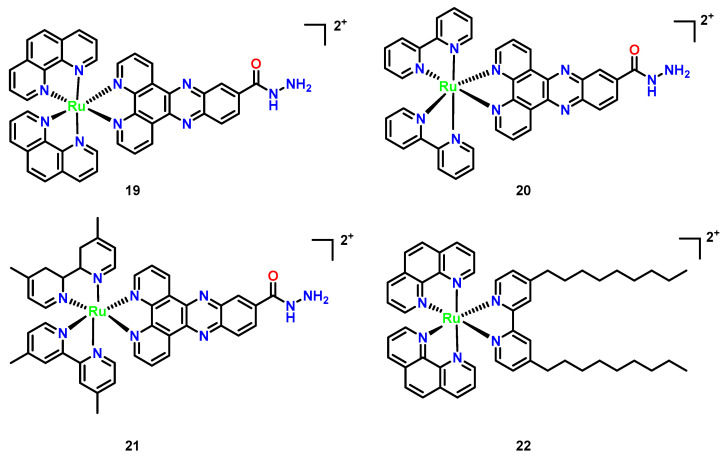
Chemical structures of octahedral polypyridyl (Complexes **19**–**22**). (**19**) [(phen)_2_Ru(dppz-carbohydrazide)]^2+^; (**20**) [(bpy)_2_Ru(dppz-carbohydrazide)]^2+^; (**21**) [(Me_2_bpy)_2_Ru(dppz-carbohydrazide)]^2+^; and (**22**) [(phen)_2_Ru(bpy-dinonyl)_2_]^2+^.

**Figure 10 microorganisms-13-01570-f010:**
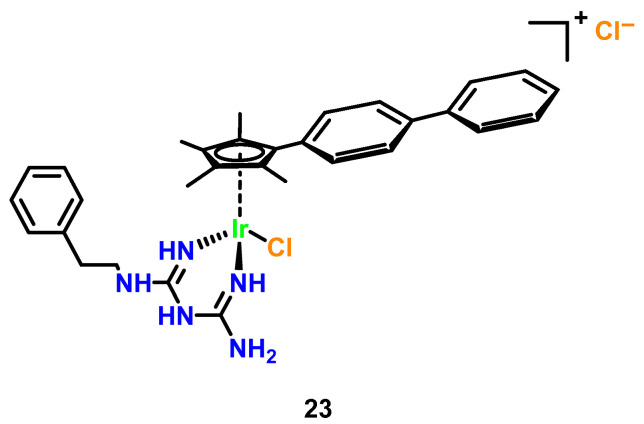
Synergistic antimicrobial iridium(III)-metformin cyclopentadienyl complex (Ir-MFC).

**Figure 11 microorganisms-13-01570-f011:**
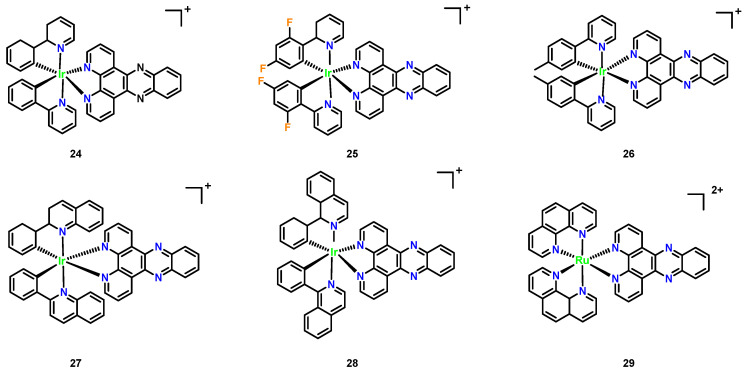
Chemical structures of polypyridyl iridium(III) complexes (Complexes **24**–**28**). (**24**) [(phenylpyridine)_2_Ir(dppz)]^+^; (**25**) [(difluorophenyl-pyridine)_2_Ir(dppz)]^+^; (**26**) [(ptpy)_2_Ir(dppz)]^+^; (**27**) [(phenylquinoline)_2_Ir(dppz)]^+^; (**28**) [(phenylisoquinoline)_2_ Ir(dppz)]^+^; and ruthenium complex (**29**) [Ru(phen)_2_(dppz)]^2+^.

**Table 1 microorganisms-13-01570-t001:** Antimicrobial spectrum and hemolytic activity (HC_50_) of Complex **26** and Complex **29**, with fluconazole as the control [45].

Organism	MIC (μg/mL)		
	26	Fluconazole	29
*C. albicans* SC5314	1	1	64
*C. albicans* G5	2	32	>64
*C. albicans* Caci17	8	>64	>64
*C. parapsilosis* ATCC2019	2	2	8
*C. krusei* ATCC6258	8	32	64
*C. neoformans* H99	4	4	4
*C. neoformans* 5-FC	4	2	4
*C. gattii* R265	2	4	>64
HC_50_	64	-	>256

**Table 2 microorganisms-13-01570-t002:** Comparative analysis of transition metal complexes: antibacterial mechanisms and efficacy.

Metal	Key Mechanisms	MIC Range (μg/mL)	Model Pathogens	Major Limitations
Silver	ROS generation, membrane disruption,	0.5–168	*P. aeruginosa*, MRSA	Mammalian cytotoxicity, efflux pump induction
Copper	DNA gyrase inhibition, Fenton-like ROS	0.5–32	*S. aureus*, *E. coil*	Narrow therapeutic window, oxidation instability
Platinum	DNA crosslinking, helical distortion	2–16	MDR-TB, CRE	Nephrotoxicity, poor solubility
Ruthenium	DNA/RNA damage, membrane penetration	1.6–16.2	*C. neoformans*, MRSA	Hemolytic effects, Gram-negative selectivity
Iridium	DNA intercalation, biofilm disruption	0.125–8	VRE, *C. albicans*	Efflux-mediated resistance (Pseudomonas)

## Data Availability

No new data were created or analyzed in this study. Data sharing is not applicable to this article.

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
