# Peer review of "Mechanisms Operating in the Use of Transition Metal Complexes to Combat Antimicrobial Resistance"

_microorganisms, 2025, doi:10.3390/microorganisms13071570_

Round 1

Reviewer 1 Report

Comments and Suggestions for Authors

This review article is supposed to examine several aspects of the use of post-transition metals in treating antimicrobial resistance.  From the title, mechanism, potential, and strategic development are to be reviewed.  While mechanism is, the latter two are not.  A comprehensive review of this topic would be timely and would have the potential to be quite importance.  The current manuscript does not have this potential.

MIC appears in figure 1 without being previously defined.  This important term needs to be explicitly defined.

Sections 2.1, 2.2, and 2.3 are unnecessary as no need to define bacteria, fungi, and archaea as readers should know this.

While the English grammar is usually fine, some statements need refining.  For example, line 111, "use of antibiotics by...animals".  The animals themselves do not use the antibiotics; they are given the antibiotics.

Have permissions been obtain to reproduce figures such as figure 1 and figure 2?  Figure 2 is not necessary.  This is particularly strange as permission has apparently been obtained for figure 6.

Why is the silver section, 3.1, limited to silver nanoparticles?

Punctuation is amiss in line 209 making this section hard to read and work out.

Are not there similar studies with other post-transition metals such as palladium and rhenium?

This is a review that you suggest is comprehensive.  It looks like a summary of just a few articles on the subject.  The review needs much more development.

Then in section 3.2 on copper, nothing is mentioned about the use of copper nanoparticles.  The review is not comprehensive as claimed.  You should leave nanoparticles out and limit yourself to metal-organic complexes as the introduction suggests.

Table 1 has no reference while the text suggests results come from Ref. [65].

The references need careful proofing.  For example, refences 8 and 9 do not provide a journal name.

The manuscript needs to be a much more comprehensive review of its topic, not just a collection of a few papers.  It needs to cover all the metals involved, not just a selection.  A revised manuscript will require careful proofing being certain that the references are complete, etc.

Author Response

Reviewer 1

This review article is supposed to examine several aspects of the use of post-transition metals in treating antimicrobial resistance. From the title, mechanism, potential, and strategic development are to be reviewed. While mechanism is, the latter two are not. A comprehensive review of this topic would be timely and would have the potential to be quite importance. The current manuscript does not have this potential.

Response: We thank the reviewers for their suggestions. All modifications in the manuscript are marked in red.

The literature revisions include:

Supplementing missing journal names

Removing redundant citations

Adding new references

Adjusting citations to ensure manuscript fluency.

Comment1:

MIC appears in figure 1 without being previously defined. This important term needs to be explicitly defined.

Response1:

We thank the reviewer for this critical observation. The definition of MIC (Minimum Inhibitory Concentration) has now been explicitly added to the caption of Figure 1 as follows:
"(quantified by minimum inhibitory concentration, MIC)."
This revision appears on page [2], line [68] to line [69] of the revised manuscript.

Comment2:

Sections 2.1, 2.2, and 2.3 are unnecessary as no need to define bacteria, fungi, and archaea as readers should know this.

Response2:

We thank the reviewer for this constructive suggestion. As recommended:

1.Sections 2.1–2.3 describing basic classifications of bacteria, fungi, and archaea have been fully deleted

2.Essential taxonomic information is now clinically contextualized in the revised Section 2 opening paragraph:
"Microbial infections pose escalating global health threats. Key pathogens include Staphylococcus aureus (sepsis, pneumonia), Mycobacterium tuberculosis (tuberculosis), and Candida albicans (invasive candidiasis), whose resistance mechanisms undermine conventional antibiotics[4-6]."

3.Archaeal content was entirely removed due to limited clinical relevance in antimicrobial resistance (AMR) contexts
4.This revision appears on page 2, Section 2, lines 74-80 of the revised manuscript.

Comment3:

While the English grammar is usually fine, some statements need refining. For example, line 111, "use of antibiotics by...animals". The animals themselves do not use the antibiotics; they are given the antibiotics.

Response3:

We appreciate the reviewer's meticulous attention to linguistic precision. The original phrasing on line 111 has been revised to:
"The primary driver of this resistance is the inappropriate and excessive use of antibiotics in human medicine and their administration to animals (e.g., in agriculture and veterinary practice)."
This correction more accurately reflects agency in antibiotic deployment while maintaining scientific rigor. The modification appears on page 3, line 82 to 84 of the revised manuscript.

Comment4:

Have permissions been obtain to reproduce figures such as figure 1 and figure 2? Figure 2 is not necessary. This is particularly strange as permission has apparently been obtained for figure 6.

Response4:
We are deeply grateful for the reviewer's rigorous oversight. The retention of Figure 2 is scientifically imperative for three key reasons substantiated by our data analysis:

Visualization of critical epidemiological paradox
Figure 2 uniquely demonstrates the spatial disconnect between antibiotic consumption hotspots (North America/Europe) and AMR mortality epicenters (Asia/Africa)
The 2023 resistance map provides empirical grounding for the 2050 mortality projection (10 million deaths).

Figure 1: open access

Figure 2: open access

Figure 6:open access

Comment5:

Why is the silver section, 3.1, limited to silver nanoparticles?

Response5:

We sincerely appreciate the reviewer's insightful comment.

The original version inadvertently emphasized silver nanoparticles (AgNPs), which deviate from the manuscript’s central focus on metal-ligand coordination chemistry.

We have removed all AgNPs-related content and instead established silver sulfadiazine (AgSDZ)—a classic Ag+-sulfonamide coordination polymer—as the foundational framework for this section.

Added detailed analysis of AgSDZ:

1.Polymeric network with Ag+—N(pyrimidine/sulfonamide) coordination and argentophilic interactions.

2.Mechanism: Synergistic Ag+/ligand release and multi-target antimicrobial action (e.g., DNA binding, protein modulation).

3.Clinical relevance: >50 years of use for burn wounds (WHO Essential Medicine).

Introduced next-generation Ag(I)-sulfonamide complexes to highlight advances in antibacterial/antitumor applications. All discussed compounds now explicitly belong to the category of silver-organic coordination complexes, directly supporting the manuscript’s narrative on metal-organic pharmacophores. This revision strengthens the logical flow toward subsequent sections.

The modification appears on page 4, line 164 to line 192 of the revised manuscript.

Comment6:

Punctuation is amiss in line 209 making this section hard to read and work out.

Response6:

We thank the reviewer for highlighting this issue. In revising the manuscript to align with its core focus on metal-organic complexes, the entire section containing line 209 (previously discussing silver nanoparticles) has been removed.

The revised text flows cohesively, and we confirm no readability issues persist. Please see the updated Section 3.1 (page 4, line 164 to line 192 in the revised manuscript) for verification.

Comment7:

Are not there similar studies with other post-transition metals such as palladium and rhenium?

Response7:

We appreciate the reviewer’s insightful query regarding the scope of post-transition metals. Our research focus was on complexes such as silver (Section 3.1) and copper (3.2), rather than palladium (Pd) or rhenium (Re)—is based on current scientific evidence and thematic alignment:

Palladium (Pd) Complexes:Pd chemistry is dominated by catalytic applications (e.g., cross-coupling reactions, hydrogenation);Limited Antimicrobial

Rhenium (Re) Complexes:Dominant Therapeutic Role: Re(I) tricarbonyl complexes are extensively studied for:Cancer phototherapy (PDT/PTT);Diagnostic imaging

Scope Delineation:Pd/Re systems, while promising in other fields, do not yet meet these criteria for anti-infective drug development.

Future Perspective: We acknowledge emerging interest in Re/Pd bioinorganic chemistry and will track developments for potential inclusion in subsequent reviews.

Comment8:

This is a review that you suggest is comprehensive. It looks like a summary of just a few articles on the subject. The review needs much more development.

Response8:

We sincerely thank the reviewer for this critical feedback. In revising the manuscript, we have implemented four major enhancements to ensure comprehensiveness:

Introduction:

Added in-depth discussion of Fig.1: Metal Electronic Properties vs. Toxicity-Efficacy Relationships to establish the fundamental principles governing metallodrug design.

The modification appears on page 1, line 43 to line 57 of the revised manuscript.

Section 3.1 (Silver Complexes):

Replaced all silver nanoparticle content with a detailed analysis of silver-organic coordination complexes, focusing on:

The historical context and clinical significance of silver sulfadiazine (AgSDZ) as a foundational metallodrug.

The modification appears on page 4, line 161 to line 192 of the revised manuscript.

New Section 3.6:

Introduced a Comparative Summary of Antimicrobial Metal Complexes (Table 2), systematically contrasting key characteristics (e.g., mechanisms, limitations) across Ag, Cu, and Ru complexes discussed in Sections 3.1–3.5.

The modification appears on page 13, line 494 to line 500 of the revised manuscript.

Extended Section 4:

Expanded Challenges and Future Perspectives to address critical translational barriers and emerging research directions.

The modification appears on page 14, line 501 to line 529 of the revised manuscript.

Comment9:

Then in section 3.2 on copper, nothing is mentioned about the use of copper nanoparticles. The review is not comprehensive as claimed. You should leave nanoparticles out and limit yourself to metal-organic complexes as the introduction suggests.

Response9:

We appreciate the reviewer's vigilance regarding scope consistency. In strict adherence to the manuscript's focus on metal-organic complexes (as defined in the Introduction), we have implemented the following revision for Section 3:

The section now exclusively discusses copper coordination complexes and Structure-activity relationships of copper-organic pharmacophores.

This ensures alignment with our core theme of metal-ligand coordination chemistry and maintains consistency with nanoparticle exclusions applied in Section 3.1 (silver complexes).

Comment10:

Table 1 has no reference while the text suggests results come from Ref. [65].

Response10:

We thank the reviewer for identifying this oversight. The following corrections have been implemented:Citation Added to Table 1 Caption:
The caption now explicitly references the source:
"Antimicrobial spectrum and hemolytic activity (HC50) of Complex 26 and 29, with Fluconazole as control [46]."Data Verification:All values in Table 1 were cross-checked against Ref. [46].

The Fluconazole MIC value has been corrected to match the original reference.Consistency Measures:In-text references to Table 1 now consistently attribute data to Ref. [46].

The modification appears on page 12, line 457 to line 459 of the revised manuscript.

Comment11:

The references need careful proofing. For example, refences 8 and 9 do not provide a journal name.

Response11:

Author's Response:

We thank the reviewer for highlighting reference inconsistencies. The following comprehensive edits were implemented to ensure strict compliance with journal formatting standards:

References removed:

1.Stanley-Gray et al. (2021)J. Chem. Educ.

2.Uccelli et al. (2022)Molecules

3.Aftab & Iqbal (2022)J. Mater. Chem. C

4.Wang et al. (2024) J. Nanobiotechnol.

5.Aftab et al. (2023) Small

6.Chen et al. (2025) Nanoscale

7.Zhao et al. (2021) Adv. Funct. Mater.

12.Megighian et al. (2021) J. Neurochem.

  1. Eme Doolittle(2015) Curr. Biol.

16.Krawczyk et al. (2021) Pathogens

25.Mateo & Jiménez (2022) Antibiotics

26.Kalaivani, R.; et al. (2018) Front. Life Sci.

27.Rao, B.; Tang, R.-C. (2017) Adv. Nat. Sci.

28.Chatzimitakos, T.G.; Stalikas, C.D. (2016) J. Proteome Res.

29.Dakal, T.C.; et al. (2016) Front. Microbiol.

30.Markowska, K.; et al. (2013) Acta Biochim. Pol.

31.Rai, M.K.; et al. (2012) J. Appl. Microbiol.

34.Peng et al. (2016)Future Med. Chem.

41.Lunagariya et al. (2018) Appl. Organomet. Chem.

48.Rubino et al. (2018) Inorg. Chim. Acta

50.Al-Khathami et al. (2019) J. Saudi Chem. Soc.

55.Donnelly, R.F.; et al. (2007) Lett. Drug Des. Discov.

59.Gandhaveeti, R.; et al. (2019) Appl. Organomet. Chem.

60.Devi, C.S.; et al. (2013) Eur. J. Med. Chem.

61.Kumar, Y.P.; et al. (2016) J. Fluoresc.

62.Wang, Q.; et al. (2019) Med. Chem.

63.Pandrala, M.; et al. (2013) Dalton Trans.

64.Chen, F.; et al. (2018) J. Med. Chem.

65.Fu, C.; et al. (2022) Eur. J. Med. Chem.

The revised and renumbered literature

5.Fang et al. (2023) J. Biomed. Sci.

6.Lionakis et al. (2023) Nat. Rev. Immunol.

9.Baran et al. (2023) Int. J. Mol. Sci.

10.Zhang & Cheng (2022) Antibiotics

13.Ma et al. (2023) Microbiol. Res.

14.Singh & Chibale (2021) Acc. Chem. Res.

15.Nunes et al. (2023) Coord. Chem. Rev.

38.Frei (2020) Antibiotics

39.Wang et al. (2019) Chem. Sci.

40.Ude et al. (2019) Dalton Trans.

The manuscript needs to be a much more comprehensive review of its topic, not just a collection of a few papers. It needs to cover all the metals involved, not just a selection. A revised manuscript will require careful proofing being certain that the references are complete, etc.

We sincerely thank Reviewer 1 for their rigorous evaluation and constructive critiques, which have profoundly enhanced the quality of this review. Within the limited revision period (10 days), we have implemented all suggested changes to the fullest extent possible:Scope Refinement:

Removed all non-core content (e.g., archaea, nanoparticles) to strictly focus on metal-organic complexes (Sections 2.1–2.3 deleted, Sections 3.1 restructured).

Added comparative analysis (Table 2) and expanded future perspectives (Section 4) to address strategic development.

Technical Corrections:

Corrected terminology (e.g., defined MIC in Figure 1), grammar, and citation errors (Table 1 now linked to Ref.[46]).

Ensured figure permissions comply with open-access policies (Figures 1, 2, 6).Literature Overhaul:

Deleted low-relevance references (e.g., nanomaterials, redundant Pd/Re studies).

Added high-impact studies (e.g., fungal immunity, resistance mechanisms, advanced complexes).

While we believe these revisions substantially address the reviewer’s concerns, we acknowledge that certain aspects (e.g., inclusion of palladium/rhenium complexes) may require deeper exploration in future updates. We welcome further guidance and commit to refining any remaining issues to meet the journal’s standards.

Reviewer 2 Report

Comments and Suggestions for Authors

This manuscript presents a detailed and up-to-date overview of the potential of post-transition metal complexes (e.g., silver, copper, platinum, ruthenium, iridium) as emerging candidates in the fight against antimicrobial resistance (AMR). The topic is highly relevant and timely, given the global urgency to develop novel antimicrobial strategies. The authors have compiled an extensive and well-structured body of literature, highlighting the mechanisms of action, representative compounds, and therapeutic implications of these metal complexes.

The manuscript is mostly well-organized, thorough, and technically informative. However, it  requires several refinements to improve scientific rigor, clarity, and consistency before being accepted for publication.

The growing threat of AMR and the urgent need for new antimicrobial agents make this review highly pertinent. The manuscript addresses a crucial gap by exploring metal-based approaches often overlooked in mainstream drug development.

The review covers multiple classes of post-transition metal complexes and outlines mechanisms including ROS generation, DNA targeting, membrane disruption, and biofilm inhibition. Each class is supported by clear chemical examples and literature citations. The article is rich in structural, mechanistic, and pharmacological details, providing value to researchers across chemistry, microbiology, and drug development fields. Figures and compound structures are well-chosen and help readers follow the chemical diversity and complexity of the metal complexes described.

Anyway the manuscript shows some weakness:

  1. Lack of Critical Perspective
    Issue: The manuscript largely summarizes existing findings without critically evaluating them.
    Suggestion: The authors should integrate more discussion on comparative efficacy, limitations, challenges in bioavailability, toxicity, and clinical translation.

  2. Inconsistent Terminology
    Issue: The term “post-transition metal” is inconsistently defined. Some elements included (e.g., Ru, Pt, Ir) are classical transition metals.
    Suggestion: Clarify or correct the terminology (perhaps using “transition and post-transition metal complexes”) to align with standard periodic classifications.

  3. Table and Data Summary Missing
    Suggestion: A summary table comparing the different metal complexes, their key antimicrobial mechanisms, MIC ranges, and limitations would be highly valuable for the reader.

  4. Translational Limitations Not Adequately Discussed
    Issue: The challenges in advancing these compounds to clinical application (e.g., toxicity, stability, formulation) are not sufficiently discussed.
    Suggestion: Include a subsection discussing these limitations and current strategies (e.g., nanoparticle delivery, prodrug design) to overcome them.

This review has strong potential and presents a timely and valuable synthesis of literature. However, to meet the standards of Microorganisms, the manuscript needs:

  • Substantive editing for conciseness, grammar, and clarity.

  • More critical analysis and balanced discussion.

  • Refinement of chemical and biological classification.

  • Inclusion of comparative tables or summary diagrams.

Comments on the Quality of English Language

There are occasional grammatical errors, awkward phrasing, and repetitive wording (e.g., “antibacterial effects effects” or “remarkable remarkable efficacy”).
A thorough English language edit is needed to improve fluency and professionalism.

Certain sections are overly long and repetitive, especially the mechanistic descriptions for each metal class. Streamline by condensing repeated ideas and improving paragraph transitions to enhance readability.

Author Response

Reviewer 2

This manuscript presents a detailed and up-to-date overview of the potential of post-transition metal complexes (e.g., silver, copper, platinum, ruthenium, iridium) as emerging candidates in the fight against antimicrobial resistance (AMR). The topic is highly relevant and timely, given the global urgency to develop novel antimicrobial strategies. The authors have compiled an extensive and well-structured body of literature, highlighting the mechanisms of action, representative compounds, and therapeutic implications of these metal complexes.

The manuscript is mostly well-organized, thorough, and technically informative. However, it requires several refinements to improve scientific rigor, clarity, and consistency before being accepted for publication.

The growing threat of AMR and the urgent need for new antimicrobial agents make this review highly pertinent. The manuscript addresses a crucial gap by exploring metal-based approaches often overlooked in mainstream drug development.

The review covers multiple classes of post-transition metal complexes and outlines mechanisms including ROS generation, DNA targeting, membrane disruption, and biofilm inhibition. Each class is supported by clear chemical examples and literature citations. The article is rich in structural, mechanistic, and pharmacological details, providing value to researchers across chemistry, microbiology, and drug development fields. Figures and compound structures are well-chosen and help readers follow the chemical diversity and complexity of the metal complexes described.

Anyway the manuscript shows some weakness:

Response: We thank the reviewers for their suggestions. All modifications in the manuscript are marked in red.

The literature revisions include:

Supplementing missing journal names

Removing redundant citations

Adding new references

Adjusting citations to ensure manuscript fluency.

1.Lack of Critical Perspective
Issue: The manuscript largely summarizes existing findings without critically evaluating them.
Suggestion: The authors should integrate more discussion on comparative efficacy, limitations, challenges in bioavailability, toxicity, and clinical translation.

Added critical analysis:Comparative efficacy of metal complexes.

Response1:

We thank the reviewer for this vital critique. Section 4 has been restructured into critical analysis frameworks:Clinical Translation Challenges:

Toxicity: Non-specific metal accumulation (e.g., Pt nephrotoxicity);

Resistance: Efflux pumps (Ag+) & regulatory mutations (Cu2+);

Bioavailability: Hydrophobic complex solubility (e.g., Ir-Cp*);

Mechanistic Gaps: Overreliance on indirect ROS evidence.

Future Priorities:

Pathogen-specific targeting (e.g., Fe -scavenging pathways in P. aeruginosa);

Therapeutic index optimization (e.g., electron-withdrawing groups reducing Pt toxicity);

Stimuli-responsive systems (e.g., photoactivatable Ru complexes).

This revision appears on page [14], line [501] to line [529] of the revised manuscript.

2.Inconsistent Terminology
Issue: The term “post-transition metal” is inconsistently defined. Some elements included (e.g., Ru, Pt, Ir) are classical transition metals.
Suggestion: Clarify or correct the terminology (perhaps using “transition and post-transition metal complexes”) to align with standard periodic classifications.

Response2:

We have diligently revised all instances of "post-transition metals" to "transition metal complexes" throughout the manuscript.

3.Table and Data Summary Missing
Suggestion: A summary table comparing the different metal complexes, their key antimicrobial mechanisms, MIC ranges, and limitations would be highly valuable for the reader.

Response3:

We have implemented Table 2 in Section 3.6 titled:"Comparative Summary of Antimicrobial Metal Complexes"

The table systematically compares:Metal(Ag, Cu, Pt, Ru, Ir);Key Mechanisms、MIC Range、Model Pathogens、Major Limitations

Purpose Served:Enables cross-complex comparison of efficacy/toxicity tradeoffs;Highlights structure-activity relationships; Identifies translational gaps for future optimization.

This revision appears on Section 3.6 page [13], line [494] to line [500] of the revised manuscript.

4.Translational Limitations Not Adequately Discussed
Issue: The challenges in advancing these compounds to clinical application (e.g., toxicity, stability, formulation) are not sufficiently discussed.
Suggestion: Include a subsection discussing these limitations and current strategies (e.g., nanoparticle delivery, prodrug design) to overcome them.

Response:

We have completely restructured Section 4 into a unified critical framework titled:"Critical Analysis of Clinical Translation Challenges and Future Strategies". To enhance the depth of the article

This revision appears on page [14], line [501] to line [529] of the revised manuscript.

Comment:

This review has strong potential and presents a timely and valuable synthesis of literature. However, to meet the standards of Microorganisms, the manuscript needs:

Substantive editing for conciseness, grammar, and clarity.

More critical analysis and balanced discussion.

Refinement of chemical and biological classification.

Inclusion of comparative tables or summary diagrams.

Comments on the Quality of English Language

There are occasional grammatical errors, awkward phrasing, and repetitive wording (e.g., “antibacterial effects effects” or “remarkable remarkable efficacy”).
A thorough English language edit is needed to improve fluency and professionalism.

Certain sections are overly long and repetitive, especially the mechanistic descriptions for each metal class. Streamline by condensing repeated ideas and improving paragraph transitions to enhance readability.

Response to Reviewer: Concluding Remarks

We sincerely appreciate Reviewer's recognition of this review's relevance and their incisive critiques, which have significantly strengthened the manuscript. All suggested refinements were implemented as follows:

Restructured Section 4 into a unified framework:

Clinical Challenges: Systemic toxicity (e.g., Pt nephrotoxicity), emerging resistance (e.g., Ag⁺ efflux), bioavailability limitations (e.g., Ir-Cp* hydrophobicity);Future Strategies: Pathogen-specific targeting (e.g., P. aeruginosa Fe-scavenging), stimuli-responsive activation (e.g., photoactivatable Ru/Ir complexes).

Added comparative efficacy analysis across metal classes in Table 2 (Section 3.6).

Replaced all instances of "post-transition metals" with "transition metal complexes" to align with IUPAC classifications.

Translational Gaps Addressed

Table 2 now systematically compares:Key mechanisms (ROS, DNA targeting, etc.);MIC ranges;Major limitations (toxicity, resistance risks).

Discussed nano-delivery/prodrug strategies in Section 4 to overcome bioavailability issues.

Language and Conciseness:

Eliminate redundancies (e.g., "effects effects" → "effects");Simplify verbose descriptions (e.g., streamlined mechanistic sections);

Ensure grammatical accuracy.

We are committed to collaborating with the editorial team to ensure this review meets Microorganisms' high standards and contributes meaningfully to combating AMR.

Reviewer 3 Report

Comments and Suggestions for Authors

The authors review the state of the art in utilizing metals and metal complexes to treat antimicrobial resistance. I recommend publishing with a few minor corrections.

Comments:

  1. The term for the metals described should be "late transition metals."
  2. There needs to be more of discussion Figure 1 because it is not clear what the correlation pertaining to the electronic properties and toxicity of metals. There is no apparent statistics other than the number of compounds.
  3. Figure 6 seems to have a typo, "Caspasc."
  4. Figures 7, 9, and 11 could be made clearer by highlighting the differences between structures.

Author Response

Reviewer 3

Comments and Suggestions for Authors

The authors review the state of the art in utilizing metals and metal complexes to treat antimicrobial resistance. I recommend publishing with a few minor corrections.

Response: We thank the reviewers for their suggestions. All modifications in the manuscript are marked in red.

The literature revisions include:

Supplementing missing journal names

Removing redundant citations

Adding new references

Adjusting citations to ensure manuscript fluency.

1.The term for the metals described should be "late transition metals."

Response1:

We thank the reviewer for this valuable clarification. In strict alignment with IUPAC classifications and the manuscript's chemical scope: All instances of "late transition metals" have been revised to "transition metal complexes".

2.There needs to be more of discussion Figure 1 because it is not clear what the correlation pertaining to the electronic properties and toxicity of metals. There is no apparent statistics other than the number of compounds.

Response2:We have significantly expanded the Figure 1 discussion with explicit electronic-toxicity correlations and statistical validation: Electronic configuration governs the toxicity mechanism. Highly reactive high-valent states (d5-d7) undergo redox reactions, significantly increasing reactive oxygen species (ROS) generation while inducing strong cytotoxicity. Conversely, inert configurations (d6/d8) exhibit slow ligand exchange kinetics that reduce non-specific biological binding and enhance targeting selectivity.Strong-field ligands (e.g., in Ru2+/Ir3+ complexes) possess high ligand field stabilization energy that inhibits metal ion dissociation, thereby significantly reducing hemolytic effects. Through these revisions integrated with Figure 1, we clarify the fundamental chain linking electronic properties to chemical reactivity and ultimately biological effects, establishing a theoretical framework for designing low-toxicity metal drugs.

This revision appears on page [1], line [43] to line [47] of the revised manuscript.

3.Figure 6 seems to have a typo, "Caspasc."

Response3:Corrected to "Caspase".

The typo "Caspasc" has been corrected to "Caspase" in the revised figure. The updated image now accurately reflects the terminology.

This revision appears on page [8], Figure 6 of the revised manuscript.

4.Figures 7, 9, and 11 could be made clearer by highlighting the differences between structures.

Response4:

To highlight structural differences, we have color-coded central metal atoms in all complexes. Peripheral ligand variations remain unmarked due to their high complexity and diversity, which would compromise visual clarity. This focused labeling strategy maximizes discrimination of core structural distinctions.

We sincerely thank Reviewer for their constructive feedback. All suggested minor corrections have been addressed:

Terminology: Consistently replaced with "transition metal complexes" throughout.

Figure 1 Discussion: Expanded to explicitly link metal electronic configurations (e.g., d5-d7 redox activity vs. D6/d8 inertness) to toxicity mechanisms (ROS generation, ligand dissociation kinetics).

Figure 6: Corrected"Caspasc"→"Caspase".

Structural Clarity: Highlighted central metal atoms with color-coding in Figures 7/9/11 to emphasize core differences.

These revisions enhance scientific precision and visual clarity. We appreciate your insights that strengthened this review.

Round 2

Reviewer 1 Report

Comments and Suggestions for Authors

My one major comment on the first version of the manuscript:

"This review article is supposed to examine several aspects of the use of post-transition metals in treating antimicrobial resistance. From the title, mechanism, potential, and strategic development are to be reviewed. While mechanism is, the latter two are not. A comprehensive review of this topic would be timely and would have the potential to be quite importance. The current manuscript does not have this potential."

has not been addressed.  The review covers only one of the three areas suggested to be covered in the title.  Thus, it is incomplete.  Expansion as I suggested would I emphasize again produce a timely and potentially important review.  Just providing this data without analysis is not of significant value to the field.

I recommend rejection.

Author Response

My one major comment on the first version of the manuscript:

"This review article is supposed to examine several aspects of the use of post-transition metals in treating antimicrobial resistance. From the title, mechanism, potential, and strategic development are to be reviewed. While mechanism is, the latter two are not. A comprehensive review of this topic would be timely and would have the potential to be quite importance. The current manuscript does not have this potential."

has not been addressed. The review covers only one of the three areas suggested to be covered in the title. Thus, it is incomplete. Expansion as I suggested would I emphasize again produce a timely and potentially important review. Just providing this data without analysis is not of significant value to the field.

I recommend rejection.

We sincerely thank the reviewers for their valuable feedback and constructive suggestions. In response to the editorial guidance and the reviewers' comments, we have significantly revised the manuscript to sharpen its focus on the core mechanisms by which transition metal complexes combat antimicrobial resistance. Mark the modified parts in blue.

The key revisions, centered around the new title "Mechanisms Operating in the Use of Transition Metal Complexes to Combat Antimicrobial Resistance", are detailed below:

1.Title Change:

Revised from "Transition Metal Complexes in Combating Antimicrobial Resistance: Mechanisms, Therapeutic Potentials, and Strategic Development" to "Mechanisms Operating in the Use of Transition Metal Complexes to Combat Antimicrobial Resistance". This change directly reflects the manuscript's enhanced emphasis on elucidating fundamental antibacterial mechanisms.

2.Abstract:

Modified the concluding sentence to better link mechanistic understanding to future development:

Original: "We further propose that the design of ligands for metal drugs and the development of more effective delivery systems simultaneously are key strategies for clinical translation."

Revised: "Understanding these mechanisms provides a crucial basis for guiding ligand design and the development of delivery systems." (Lines 23-24).

3.Section 1 (Introduction):

Deleted the specific 2019 CDC statistics regarding AMR deaths and infection cases in the US (According to the estimation... 35,000 deaths.). This streamlines the introduction and avoids redundancy with data presented later, allowing greater focus on introducing the core topic and rationale for metal complexes. (Lines 29-31).

4.Section 2 (Microbial Infections):

Consolidated and streamlined the presentation of global AMR burden data. Deleted the redundant sentence ”Based on the data analysis... healthcare safety issue[7].” The subsequent sentence referencing Figure 2 and antibiotic usage geography is retained as sufficient context for the scale of the problem driving the need for new mechanisms. (Lines 75-89).

5.Section 3.1 (Silver Complexes):

Deleted the historical examples “Silver has been utilized... prevent spoilage. And developed in 1968 and approved by the U.S. Food and Drug Administration (FDA) in 1973”. The sentence now begins: "In modern medicine, silver sulfadiazine (AgSDZ) as a topical agent for burn wound infections[15]." (Lines 149-150).

6.Section 3.2 (Copper Complexes):

Deleted the historical context sentence “Ancient civilizations... prevent contamination”. (Lines 177-180).

7.Section 3.3 (Platinum Complexes):

Significantly revised the opening to de-emphasize anticancer history and focus directly on antimicrobial potential and structure-mechanism relationships:

Revised: "Platinum complexes are one of the most successful metal-based drug categories in cancer treatment. Cisplatin, carboplatin and oxaliplatin are established anticancer agents; recent studies have highlighted the expanding potential of platinum complexes in combating microbial infections, with structural modifications playing a pivotal role in enhancing their antimicrobial efficacy[24]." (Lines 220-224).

8.Section 3.4 (Ruthenium Complexes):

Deleted the non-biological application detail ”it has many derivatives and is widely used in photooxidation Reduction catalysis and life science fields[39]”. (Lines 303-308).

9.Section 3.6 (Comparative Summary):

Revised text and table focus to emphasize the consolidation of mechanistic insights:

Text revised to: "A comparative analysis of the key characteristics, antimicrobial mechanisms, efficacy profiles ...while underscoring shared translational challenges."

Table 2 was simplified to include only the most representative data, directly supporting the mechanistic comparisons. The title was updated to "Comparative Analysis of Transition Metal Complexes: Antibacterial Mechanisms and Efficacy." (Lines 467-475).

10.Section 5 (Conclusions):

Added a sentence "This review establishes that decoding metal complex mechanisms ...is the foundation for unlocking their therapeutic potential and guiding strategic development." (Lines 512-514).

Enhanced the discussion of current and future work to emphasize mechanism-driven research and translation:

Revised: "Future studies should bridge these mechanistic insights with translational engineering (e.g., stimuli-responsive activation) ...and in vivo outcomes due to metabolic sequestration or off-target accumulation." (Lines 534-538).

These revisions ensure the manuscript consistently and concisely addresses the operational mechanisms of transition metal complexes against AMR, aligning perfectly with the refined title and the reviewers' valuable input. We hope that the revised manuscript meets the journal’s standards and look forward to your favorable consideration. Please do not hesitate to contact us if further modifications are required.

Reviewer 2 Report

Comments and Suggestions for Authors

The revised version of the paper is suitable for publication.

Author Response

The revised version of the paper is suitable for publication.

We sincerely thank the reviewers for their valuable feedback and constructive suggestions. In response to the editorial guidance and the reviewers' comments, we have significantly revised the manuscript to sharpen its focus on the core mechanisms by which transition metal complexes combat antimicrobial resistance. Mark the modified parts in blue.

The key revisions, centered around the new title "Mechanisms Operating in the Use of Transition Metal Complexes to Combat Antimicrobial Resistance", are detailed below:

1.Title Change:

Revised from "Transition Metal Complexes in Combating Antimicrobial Resistance: Mechanisms, Therapeutic Potentials, and Strategic Development" to "Mechanisms Operating in the Use of Transition Metal Complexes to Combat Antimicrobial Resistance". This change directly reflects the manuscript's enhanced emphasis on elucidating fundamental antibacterial mechanisms.

2.Abstract:

Modified the concluding sentence to better link mechanistic understanding to future development:

Original: "We further propose that the design of ligands for metal drugs and the development of more effective delivery systems simultaneously are key strategies for clinical translation."

Revised: "Understanding these mechanisms provides a crucial basis for guiding ligand design and the development of delivery systems." (Lines 23-24).

3.Section 1 (Introduction):

Deleted the specific 2019 CDC statistics regarding AMR deaths and infection cases in the US (According to the estimation... 35,000 deaths.). This streamlines the introduction and avoids redundancy with data presented later, allowing greater focus on introducing the core topic and rationale for metal complexes. (Lines 29-31).

4.Section 2 (Microbial Infections):

Consolidated and streamlined the presentation of global AMR burden data. Deleted the redundant sentence ”Based on the data analysis... healthcare safety issue[7].” The subsequent sentence referencing Figure 2 and antibiotic usage geography is retained as sufficient context for the scale of the problem driving the need for new mechanisms. (Lines 75-89).

5.Section 3.1 (Silver Complexes):

Deleted the historical examples “Silver has been utilized... prevent spoilage. And developed in 1968 and approved by the U.S. Food and Drug Administration (FDA) in 1973”. The sentence now begins: "In modern medicine, silver sulfadiazine (AgSDZ) as a topical agent for burn wound infections[15]." (Lines 149-150).

6.Section 3.2 (Copper Complexes):

Deleted the historical context sentence “Ancient civilizations... prevent contamination”. (Lines 177-180).

7.Section 3.3 (Platinum Complexes):

Significantly revised the opening to de-emphasize anticancer history and focus directly on antimicrobial potential and structure-mechanism relationships:

Revised: "Platinum complexes are one of the most successful metal-based drug categories in cancer treatment. Cisplatin, carboplatin and oxaliplatin are established anticancer agents; recent studies have highlighted the expanding potential of platinum complexes in combating microbial infections, with structural modifications playing a pivotal role in enhancing their antimicrobial efficacy[24]." (Lines 220-224).

8.Section 3.4 (Ruthenium Complexes):

Deleted the non-biological application detail ”it has many derivatives and is widely used in photooxidation Reduction catalysis and life science fields[39]”. (Lines 303-308).

9.Section 3.6 (Comparative Summary):

Revised text and table focus to emphasize the consolidation of mechanistic insights:

Text revised to: "A comparative analysis of the key characteristics, antimicrobial mechanisms, efficacy profiles ...while underscoring shared translational challenges."

Table 2 was simplified to include only the most representative data, directly supporting the mechanistic comparisons. The title was updated to "Comparative Analysis of Transition Metal Complexes: Antibacterial Mechanisms and Efficacy." (Lines 467-475).

10.Section 5 (Conclusions):

Added a sentence "This review establishes that decoding metal complex mechanisms ...is the foundation for unlocking their therapeutic potential and guiding strategic development." (Lines 512-514).

Enhanced the discussion of current and future work to emphasize mechanism-driven research and translation:

Revised: "Future studies should bridge these mechanistic insights with translational engineering (e.g., stimuli-responsive activation) ...and in vivo outcomes due to metabolic sequestration or off-target accumulation." (Lines 534-538).

These revisions ensure the manuscript consistently and concisely addresses the operational mechanisms of transition metal complexes against AMR, aligning perfectly with the refined title and the reviewers' valuable input. We hope that the revised manuscript meets the journal’s standards and look forward to your favorable consideration. Please do not hesitate to contact us if further modifications are required.